# Synthesis and Inhibitory Activity of Machaeridiol-Based Novel Anti-MRSA and Anti-VRE Compounds and Their Profiling for Cancer-Related Signaling Pathways

**DOI:** 10.3390/molecules27196604

**Published:** 2022-10-05

**Authors:** Mallika Kumarihamy, Siddharth Tripathi, Premalatha Balachandran, Bharathi Avula, Jianping Zhao, Mei Wang, Maria M. Bennett, Jin Zhang, Mary A. Carr, K. Michael Lovell, Ocean I. Wellington, Mary E. Marquart, N. P. Dhammika Nanayakkara, Ilias Muhammad

**Affiliations:** 1National Center for Natural Products Research, Research Institute of Pharmaceutical Sciences, School of Pharmacy, University of Mississippi, University City, MS 38677, USA; 2Natural Products Utilization Research Unit, Agricultural Research Service, USA Department of Agriculture, University City, MS 38677, USA; 3Department of Cell and Molecular Biology, University of Mississippi Medical Center, 2500 North State Street, Jackson, MS 39216, USA

**Keywords:** machaeridiol analogs, synthesis, antibacterial, MRSA, VRE, in vivo nosocomial MRSA assay, anticancer, transcription factors

## Abstract

Three unique 5,6-*seco*-hexahydrodibenzopyrans (*seco*-HHDBP) machaeridiols A–C, reported previously from *Machaerium* Pers., have displayed potent activities against methicillin-resistant *Staphylococcus aureus* (MRSA), vancomycin-resistant *Enterococcus faecium,* and *E*. *faecalis* (VRE). In order to enrich the pipeline of natural product-derived antimicrobial compounds, a series of novel machaeridiol-based analogs (**1**–**17**) were prepared by coupling stemofuran, pinosylvin, and resveratrol legends with monoterpene units *R*-(−)-*α*-phellandrene, (−)-*p*-mentha-2,8-diene-1-ol, and geraniol, and their inhibitory activities were profiled against MRSA ATCC 1708, VRE ATCC 700221, and cancer signaling pathways. Compounds **5** and **11** showed strong in vitro activities with MIC values of 2.5 μg/mL and 1.25 μg/mL against MRSA, respectively, and 2.50 μg/mL against VRE, while geranyl analog **14** was found to be moderately active (MIC 5 μg/mL). The reduction of the double bonds of the monoterpene unit of compound **5** resulted in **17**, which had the same antibacterial potency (MIC 1.25 μg/mL and 2.50 μg/mL) as its parent, **5**. Furthermore, a combination study between *seco*-HHDBP **17** and HHDBP machaeriol C displayed a synergistic effect with a fractional inhibitory concentrations (FIC) value of 0.5 against MRSA, showing a four-fold decrease in the MIC values of both **17** and machaeriol C, while no such effect was observed between vancomycin and **17**. Compounds **11** and **17** were further tested in vivo against nosocomial MRSA at a single intranasal dose of 30 mg/kg in a murine model, and both compounds were not efficacious under these conditions. Finally, compounds **1**–**17** were profiled against a panel of luciferase genes that assessed the activity of complex cancer-related signaling pathways (i.e., transcription factors) using T98G glioblastoma multiforme cells. Among the compounds tested, the geranyl-substituted analog **14** exhibited strong inhibition against several signaling pathways, notably Smad, Myc, and Notch, with IC_50_ values of 2.17 μM, 1.86 μM, and 2.15 μM, respectively. In contrast, the anti-MRSA actives **5** and **17** were found to be inactive (IC_50_ > 20 μM) across the panel of these cancer-signaling pathways.

## 1. Introduction

There are currently considerable challenges with the treatment of infections caused by strains of clinically relevant bacteria that show multidrug-resistance (MDR), such as methicillin-resistant *Staphylococcus aureus* (MRSA) and *Enterococci faecalis* (VRE), as well as the recently emerged and extremely drug-resistant *Mycobacterium tuberculosis* XDR-TB [1]. The continuous gradual decline in cases of MRSA and other MDRs since 2010 was accompanied by a sudden spike due to the emergence of the SARS-CoV-2 virus. This spike represented a 34% increase in cases of MRSA, with some American states seeing an increase in cases as high as a 99% [2,3]. The national estimate for invasive MRSA incidence rates showed that one in three people carry *S. aureus* in their nose and two in 100 people carry MRSA [4]. Like MRSA, VRE infections are commonly acquired by hospitalized patients. Enterococcal infections can be lethal, particularly those caused by VRE. According to the Centers for Disease Control and Prevention (CDCP), the number of nosocomial VRE isolates increased in the United States by 20-fold between 1989 and 1993, and now such isolates are the second-to-third most common cause of nosocomial infections in the USA. Overall antibiotic resistance is on the rise, with some strains developing resistance to powerful antibiotics, such as vancomycin, that are typically used as a last resort when other efforts fail.

Because cancer patients are more likely to be infected and die from MRSA bloodstream infections (BSI), a comprehensive review was reported recently by Li et al. (2020) [5], who “estimated the global MRSA prevalence among bacteremia in patients with malignancy, and studied the predictors and mortality of cancer patients with MRSA bacteremia or BSI”. The authors observed that “the pooled prevalence of MRSA was 3% (95% CI 2–5%) among all BSIs and 44% (95% CI 32–57%) among *S. aureus* bacteremia in cancer patients” [5]. In addition, An et al. (2016) [6] found that “MRSA infection can enhance metastasis ability of A549 cell (lung cancer) and increase matrix metalloproteinase (MMP2 and MMP9) expressions in MRSA infected A549 cell, and concluded that MRSA infection can enhance NSCLC cell metastasis by up-regulating TLR4/MyD88 signaling”. Considering the urgency of newly emerging MRSA infection due to the COVID-19 pandemic and the BSI of cancer patients, novel antibacterial leads that focus on the dual effects of anti-MRSA and anticancer approaches from natural and synthetic sources are urgently warranted.

Based on our ongoing effort to uncover antimicrobial agents from natural products from higher plants, hexahydrodibenzopyran (HHDBP)-type phytocannabinoids showed promising activity against MRSA and VRE. The examples of these phytocannabinoids are machaeriols A–D and machaeridiols A–C (Figure 1), isolated from *Machaerium* Pers. (Rimachi 12161), which are very unique and rare in higher plants [7,8,9]. HHDBP occurs in chemotaxonomically unrelated sources, such as liverworts, which are non-vascular plants referred to as Marchantiophyta, and an aralkyl analogue of Δ^9^-tetrahydrocannabinol (THC), perrottetinen, was previously reported from the liverwort *Radula perrottetii* [10,11]. In addition, two Leguminoseae species, *Desmodium canum* and *Sophora tetraptera*, have yielded aralkyl phytocannabinoids [12,13]. However, we previously reported the significance of the chemistry of HHDBP or 5,6-*seco*-HHDBP (machaeriols and machaeridiols) and their antimicrobial activities, especially against MRSA, VRE, and Gram-negative bacteria [9]. Interestingly, the structural resemblance between machaeridiol and cannabidiol (CBD) (Figure 1) reflected their similarity in antimicrobial activities against MRSA, and machaeridiols A–C were found to be more potent than cannabidiol (CBD) and vancomycin [9,14].

In this paper, we report the synthesis of a series of new and diverse stemofuran-, pinosylvin-, and resveratol-type aralkyl compounds (**1**–**11, 15**–**17**) and geraniol-substituted pinosylvin analogs, such as cannabigerol (CBG) analogs (**12**–**14)**, and profile their antimicrobial and anticancer activities. Compounds **5** and **11** showed potent activity against MRSA and VRE, while compound **14** exhibited moderate anti-MRSA activity with strong anticancer activity against a panel of transcription factors. The reduction of the double bonds of compound **5** yielded two diastereomers and the major diastereomer **17** showed no change in its antibacterial activity. The total synthesis and antibacterial activities of compounds **1**–**17** toward MRSA and VRE, as well as their anticancer activities against luciferase transporter genes, are reported in this paper.

## 2. Results

Seven phytocannabinoids with HHDBP and *seco*-HHDBP skeleta, machaeriols A–D and machaeridiols A–C, were previously isolated from *Macherium* Pers., of which the machaeriol chemotype possesses a unique pharmacophore with 10a*S* absolute configuration, an opposite configuration to 10a*R* of THC (tetrahydrocannabinol) / HHC (hexahydrocannabinol) [8]. In addition, the configuration of the machaeridiol chemotype at 1*S*, 3*S*, and 4*S* positions is opposite to those at 3*R* and 4*R* positions of CBD. Therefore, an unprecedented structural similarity for the HHDBP skeleton was observed between machaeriol and HHC, and for the 5,6-*seco*-HHDBP skeleton between machaeridiol and dihydro-CBD. Because machaeriols A–D are analogues to *trans*-HHC, they show an enantiomeric configuration at the ring junction, established by CD and enantioselective total syntheses [15,16,17]. Similarly, the macheridiol chemotype is similar to dihydro-CBD, with the n-alkyl mioety replaced by steryl and benzofuranyl forms. [18] The pseudo-enantiomeric configuration at C-3 and C-4, compared to CBD, was established by CD studies. Based on the uniqueness of the stereogenic ring junction of the machaeridiol chemotype, the synthesis of machaeridiol B was reported [19] to involve long steps; to repeat this method would be quite challenging.

### 2.1. Synthesis of Analogs **1**–**17**

Based on 5,6-*seco*-HHDBP machaeridiols A and C (Figure 1), consisting of styryl and benzopyran legends, respectively, attached to cyclic monoterpenoid via a catechol unit [7,8,9], we introduced different monoterpene units, (*R*)-(−)-*a*-phellandrene, (−)-*p*-mentha-2,8-diene-1-ol [(1*R*,4*S*)-1-methyl-4-(prop-1-en-2-yl)cyclohex-2-enol], geraniol with benzopyran (stemofuran A), pinosylvin, and resveratrol legends to form new *trans*-benzopyan, styryl and resveratrol analogs **1**–**17** (Figure 1, Figure 2, Figure 3 and Figure 4). Catechol and monoterpens coupling reactions were carried out using the simple and efficient one-pot synthetic protocol described by Mascal et al. [20]. Stemofuran A was coupled with (*R*)-*a*-phellandrene and (−)-*p*-mentha-2,8-diene-1-ol (Figure 1) to yield compoundss **1**, **2**, and **3**–**5**, respectively. Similarly, pinosylvin was coupled with (−)-*p*-mentha-2,8-diene-1-ol and geraniol (Figure 2) to yield compounds **6**–**11** and **12**–**14**, respectively, and resveratrol was coupled with (−)-*p*-mentha-2,8-diene-1-ol (Figure 3) to yield compounds **15**–**16**. The structures of these compounds were confirmed by extensive 2D-NMR analysis and their configuration was determined by NOESY correlations. The complete hydrogenation of compound **5** with Adam’s catalyst yielded **17** and the absolute configuration at C-1 of **17** was determined by analyzing its NOESY data; it was assigned as 1*R* and named as 5-(benzofuran-2-yl)-4-(1*R*,2*S*,5*R*)-2-isopropyl-5-methylcyclohexyl)benzene-1,3-diol.

### 2.2. Antimicrobial Activities against Gram-Positive Bacteria and Fungi

The antimicrobial activities of compounds **1**–**17** were evaluated against a panel of microorganisms, including MRSA, vancomycin-resistant *Enterococci*, *Enterococcus faecium* (VRE), and *Cryptococcus neoformans*, together with anticancer activity against a panel of luciferase reporter genes of complex cancer-related signaling pathways. All compounds were initially subjected to in vitro antimicrobial primary screens to assess their IC_50_ values; compounds **4**, **6**, **9**+**10**, **12**, and **16** were found to be inactive (IC_50_ > 20 μg/mL) (Table 1). Analogs from the stemofuran A, pinosylvin, and resveratrol legends exhibited strong-to-modest IC_50_ values in the range of 0.91 μg/mL to 8.04 and 1.17 μg/mL to 3.38 μg/mL against MRSA and VRE, respectively. Representative analogs from each of these three legends, compounds **5**, **11**, **13**, **14** and **17**, were selected, based on IC_50_ values of 3.0 μg/mL or <3.0 μg/mL as a cut-off value against MRSA for MIC determination. Among the analogous of stemofuran A (**1**–**5** and **17**), pinosylvin (**6**–**14**), and resveratrol (**15**, **16**), compound **5, 11**, and **17** exhibited the most potent activity against MRSA ATCC 1708, with IC_50_/MIC values of 1.18/2.50 μg/mL, 0.91/1.25 μg/mL, and 0.95/1.25 μg/mL, respectively. In addition, these three analogs (**5, 11**, and **17**) were found to be active against *E. faecalus* ATCC 700221 (VRE) with IC_50_/MIC of 2.27/2.50 μg/mL, 1.17/2.50 μg/mL, and 2.28/2.50 μg/mL, respectively (Table 1). On the other hand, among the pinosylvin- geraniol analogs (**12**–**14**), compound **14** was found to be more active, compared to **12** and **13**, against MRSA and VRE, with MIC of 5 μg/mL.

The activities of compounds **5**, **11**, and **17** were found to be similar to those of natural products machaeridiol C (i.e., a stemofuran A type, **17**) and machaeridiol A (i.e., a pinosylvin type, **11**), with MIC values of 1.25 μg/mL to 2.5 μg/mL, and more potent than the positive drug control meropenem against MRSA and VRE.

### 2.3. Antimicrobial Combination Studies

Based on our previous investigation on the synergistic activity between machaeridiol B and machaeriol C against MRSA [9], analogs **5** and **17** (i.e., machaeridiol type analogs) were subjected to combination studies with machaeriol C. Because both compounds **17** and machaeriol C showed potent activity against MRSA (Table 1 and Table 2), they were subjected to a broth microdilution checkerboard assay [9], using an MRSA strain. The presence of machaeriol C improved the anti-MRSA activity of **17**; for example, **17** alone showed an MIC of 0.625 μg/mL, but in combination with machaeriol C, the MIC of **17** was decreased to 0.156 μg/mL. This combination study of **17** and machaeriol C displayed a synergistic effect with a fractional inhibitory concentration (FIC) value of 0.5 against MRSA, showing a four-fold decrease in MIC of **17** to 0.312 μg/mL (Table 2). On the other hand, compound **5** + machaeriol C displayed an additive effect with an FIC value of 0.75, showing a four-fold reduction of MIC for **5** from 1.25 μg/mL to 0.312 μg/mL and a two-fold reduction of MIC of macheriol C from 1.25 μg/mL to 0.625 μg/mL. However, a combination of either **5** or **17** with antibiotic vancomycin did not show any such effect against MRSA (Table 2). All checkerboard assays were run in triplicate.

### 2.4. In Vivo Antimicrobial Activity Studies agaisnt Nosocomial MRSA

Compounds **11** and **17** were evaluated in vivo against MRSA strain USA300, using a nosocomial assay protocol in a murine model. Both compounds were found to be unable to reduce bacterial loads in the nasopharynx and lung at a dose of 30 mg/kg, using ciprofloxacin as a reference standard. A single intranasal dose of compound, vehicle, or ciprofloxacin was administered 12 h following infection, and this regimen produced significant but modest results for ciprofloxacin (Table 3). Alternate concentrations, additional treatments, parental versus intranasal routes, and/or alternate treatment schedules need to be investigated to determine the optimal results for ciprofloxacin control, to better compare the efficacy of compounds **11** and **17**.

### 2.5. Activities against Cancer-Related Signaling Pathways

In order to evaluate the anticancer activities and understand the mechanism of action, compounds **1**–**17** were profiled against a panel of luciferase reporter genes that assessed the activity of complex cancer-related signaling transduction pathways, using T98G glioblastoma multiforme cells (i.e., transcription factors; TF; Figure 5). The compounds were considered inactive if their inhibition was less than 40% of the induction at the highest concentration tested (20 µM). These inactive compounds were not further advanced for a dose response assay for IC_50_ determination. Among the compounds (Table 4), the geraniol derivative of pinosylvin legend **14** exhibited strong activity against many signaling pathways (TFs), notably with IC_50_ values between 1.86 µM to 4.45 µM against the Stat, Smad, Myc, Ap-1, NF-kB, E2F, ETS, and Notch pathways. However, the MRSA active stemofurans **5** and **17** and the pinosylvin **11** analogs were found to be inactive and weakly active, respectively, against the panel of these signaling pathways.

## 3. Discussion

The synthesis of the compounds **1**–**17**, using stemofuran A, pinosylvin, and resveratrol legends, was inspired by the potent anti-MRSA and anti-VRE activities of machaeridiol A and machaeridiol C [9], CBD [21], and CBG [22] to develop a new pipeline of anti-MRSA active compounds. This appears to be the first report of compounds **1**, **2**, **4**–**12**, **14**, **15**, and **17** from either a natural or synthetic source. Compounds **3** and **16** were previously synthesized by coupling reactions similar to those described in the current study, but using a different catalyst (BF_3_·OEt_2_) [16] and a different solvent (DCM) [23], respectively. Compound **13** (amorphastibol) was reported from three *Amorpha* species [24]. Compounds **9** and **10** were isolated as a 5:1 mixture and very likely formed by an acid-catalyzed intermolecular cyclization of compound **11**, as previously reported for CBD under similar conditions [25]. Compound analogs to **6**–**8** and **11**, but without an exocyclic double bond in the monoterpene moiety, were reported from *Lindera reflexa* [26,27].

The HHDBP and 5,6-*seco*-HHDBPs, machaeriols and machaeridiols, were only reported from *Machaerium* Pers. (Rimachi 12161), despite our search for other *Machaerium* species available at the the repository of the National Center for Natural Products Research at the University of Mississippi. However, the stereo-specific total synthesis of machaeriols A–D and mechaeridiol B (Figure 1) has been reported [15,19,28]. In addition, several HHC-related machaeriol analogs were synthesized, showing a strong inhibition of tumor (breast cancer) growth by targeting VEGF-mediated angiogenesis signaling in endothelial cells and suppressing VEGF production and cancer cell growth [29]; however, to best of our knowledge, machaeridiol analogs have not been synthesized and reported for anti-MRSA or anticancer activities.

The antibacterial activities against MRSA and VRE, as well as anticancer activities toward signal transduction pathways, were assessed; compounds **5**, **11**, and **17** showed potent activities against MRSA and VRE, with MIC values in the range of 1.25 μg/mL to 2.50 μg/mL, while compound **14** exhibited moderate activity (MIC 5.0 μg/mL). It is intriguing to note that the combination of *seco*-HHDBP **17** with HHDBP machaeriol C displayed a synergistic effect against MRSA, showing a four-fold decrease in the MIC of both **17** and machaeriol C, while compound **5** showed an additive effect with machaeriol C and a four-fold decrease in the MIC of **5** (Table 2). The strong in vitro efficacy of compounds **11** and **17** against MRSA prompted investigation of their efficacy in vivo. A preliminary experiment employing a single intranasal dose of each compound, vehicle, and ciprofloxacin yielded modest efficacy results for ciprofloxacin only. Different parameters in the treatment regimen must be tested to fully determine whether compounds **11** and **17** are able to reduce bacterial loads in vivo. Because the results were modest for ciprofloxacin, it is possible that repeated treatment or systemic administration may be necessary to achieve efficacy. Although compound **17** was found to be inactive against in vivo nosocomial MRSA at a low dose using a murine model, our study facilitated a way forward for further mechanistic investigation on these new leads.

Finally, three geranyl-substituted structural analogs (**12**–**14**) of cannabigerol (CBG) were synthesized. Among these, compound **14** was found to be strongly active against the cancer signaling pathways, notably Smad, Myc, and Notch, with IC_50_ values between 1.2 μM and 4.8 μM in T98G glioblastoma multiforme cells. On the other hand, the anti-MRSA compounds **5** and **17** were found to be inactive against the panel of these transporter genes. A recent report by Morch et al. in 2021 [30] showed that Smad/TGF-β is a key player in probiotic protection against MRSA in *C. elegans*. In addition, in 2013, Choi et al. [31] reported that Stat3 induction helps host defense against MRSA pneumonia. Interestingly, the Notch pathway is activated by *S. aureus* toxins, in both in vivo and in vitro conditions [32], which suggests that compound **14** could have strong therapeutic potential against MRSA, due to its strong selective inhibition against the Notch pathway in this current study. Several reports suggested as association of MRSA with a significant increase in cancer mortality [5,6]. Thus, the inhibition of compound **14** against these cancer-signaling pathways could possibly decrease the morbidity of cancer conditions. Earlier, CBG, a close structural analog of **13**, was not only reported for anti-MRSA activity [22], but also for its strong activity in colon, breast, and oral carcinomas [33,34,35]. In addition, CBG has been regarded as a potential therapeutic agent for glioblastoma [36]. This combination of earlier reports and our observations from the current study strongly suggest that compound **14** should be explored further for several cancers, particularly glioblastomas (brain tumors), and in combination therapies against MRSA. The co-occurrence of alterations in the multiple pathways suggests that compound **14** could be a potential lead molecule for targeted and combination therapies. Certain pathways, such as the RAS signaling pathway, are altered across many different tumor types [37], and the inhibition of compound **14** against RAS (moderate activity with IC_50_ value of 6.5 µM) indicates that it could be explored further for several cancers. The cross-talk between the other pathways in response to compound **14** treatment (Stat 3, AP-1, NFkB, E2F, and ETS with IC_50_ in the range of 4 µm to 5 µm) reflect functional interactions and dependencies. Overall, as Yu et al. [38] reported in 2022, molecular targeted therapies play a key role in the treatment of various cancers, and our results could be the starting point for the further development of compound **14**.

## 4. Experimental

### 4.1. General Experimental Procedures

Optical rotation was determined by an AUTOPOLVR IV polarimeter. The 1H- and 13C NMR spectra were recorded on a Bruker Avance 400 or 500 MHz spectrometers. HMBC, HSQC, and ROESY were measured on an Agilent DD2-500 NMR spectrometer. The ESI-HRMS data was obtained by utilizing Agilent 6545 LC/Q-ToF and Model 6230 ToF (controlled by Agilent MassHunter Work Station, A.08.00) systems. All acquisitions were performed under a positive and negative ionization mode with a capillary voltage of 3000 V. Nitrogen was used as the nebulizer gas (25 psig) as well as the drying gas at 7 L/min at a drying gas temperature of 325 °C. Other parameters included sheath gas temperature, 300 °C; sheath gas flow, 7 L/min; skimmer, 65 V; Oct RF V, 750 V; and fragmentor voltage, 150 V. Ten microliters of sample were injected. Full scan mass spectra were acquired from *m/z* 100–1100. Data acquisition and processing was done using the MassHunter Workstation software (Qualitative Analysis Version B.10.00, Santa Clara, CA, USA). Accurate mass measurements of each spectrum from the data collected were obtained by means of reference ion correction using reference masses at *m/z* 121.0509 (protonated purine) and 922.0098 [protonated hexakis (1H, 1H, 3H-tetrafluoropropoxy) phosphazine or HP-921] in positive ion mode and at *m/z* 112.9856 (deprotonated trifluoroacetic acid-TFA) and 1033.9881 (TFA adducted HP-921) in negative ion mode. LC/MS data was measured using an Agilent 1290 Infinity series UHPLC instrument, coupled to an Agilent 6120 quadrupole mass spectrometer with a dual ESI and APCI interface. HPLC analysis was conducted on a Agilent technologies 1100 series with a diode array detector with semi preparative RP-HPLC (column: Luna C18(2) 10 µ, 250 × 10 mm; detector; UV 254 nm), using MeCN-H_2_O as the solvent. TLC analysis was carried out using analytical silica gel 60 PF254 and pre-coated alumina plates (Merck, Rahway, NJ, USA; 0.25 mm thick). The plates were visualized under a UV lamp (254 nm) and sprayed with an anisaldehyde reagent, followed by heating.

### 4.2. Purification of Analogs by Centrifugal Preparative TLC and HPLC

Compounds **1**–**17** were separated from the reaction mixtures by customized centrifugal chromatography (Spin Chromatography system) [39], and compounds **2**–**4**, **6**, **8**, **9**+**10**, and **15**–**17** were further re-purified (>95%) by semi preparative RP-HPLC (column: Luna C18(2. 10 µ, 250 × 10 mm; detector; UV 254 nm), using MeCN-H_2_O as the solvent.

### 4.3. Synthesis of Compounds **1** and **2**

(*R*)-(−)-*α*-Phellandrene (65.39 mg, 0.48 mmoL, >95%, Sigma Aldrich, St. Louis, MO, USA) was added to a mixture of stemofuran A (100 mg, 0.44 mmoL) and *p*-toluenesulfonic acid monohydrate (PTSA, 25.17 mg, 0.13 mmol) in benzene (2.5 mL) at room temperature (RT) and the reaction mixture was stirred for 1 h [20]. The reaction was quenched with aqueous NaHCO_3_ and extracted with CH_2_Cl_2_. The solvent was removed in vacuo and the residue (200 mg) was purified by CPTLC (2 mm silica gel P254 disc, 45 RPM), and elution with EtOAc in hexane (1–20%) yielded compound **1** (50 mg, 32%) and a fraction containing **2** (15 mg, 9%). The fraction with **2** was further purified by semipreparative RP-HPLC, using 80% MeCN-H_2_O as the mobile phase to afford pure compound **2** (5 mg, 3%).

#### 4.3.1. Compound **1**

(1′*R*,2′*R*)-4-(Benzofuran-2-yl)-2′-isopropyl-5′-methyl-1′,2′,3′,4′-tetrahydro-[1,1′-bihenyl]-2,6-diol; Gum, [α]^26^_D_ +188 (c 0.33, CDCl_3_),^1^H NMR (500 MHz, CDCl_3_) δ 7.58 (dd, *J* = 7.6, 1.0 Hz, 1H), 7.52 (d, *J* = 7.6 Hz, 1H), 7.31–7.23 (m, 2H), 6.94 (s, 1H), 6.35 (bs, 1H), 5.58 (s, 1H), 5.20 (bs, 1H), 3.96 (m, 1H), 2.22–2.16 (m, 2H), 1.85 (m, 1H), 1.84 (s, 3H), 1.72 (m, 1H), 1.68 (m, 1H), 1.46 (m, 1H), 0.94 (d, 6.8 Hz, 3H), 0.89 (d, 6.8 Hz, 3H). ^13^C NMR (126 MHz, CDCl_3_) δ 155.4, 154.8, 140.8, 129.9, 129.2, 124.2, 124.2, 122.9, 120.9, 117.8, 111.1, 101.4, 43.6, 35.7, 30.7, 28.0, 23.7, 23.1, 21.7, 16.4. ESIMS: *m*/*z* calcd for C_24_H_27_O_3_ [M+H]^+^: 363.19; found: 363.20. HRMS: *m*/*z* calcd for C_24_H_25_O_3_ [M − H]^−^: 361.1809; found: 361.1809.

#### 4.3.2. Compound **2**

(1′*S*,2′*S*)-6-(Benzofuran-2-yl)-2′-isopropyl-5′-methyl-1′,2′,3′,4′-tetrahydro-[1,1′-biphenyl]-2,6-diol; Gum, [α]^26^_D_ +46 (c 0.33, CDCl_3_);^1^H NMR (500 MHz, CDCl_3_) δ 7.60 (dd, J = 7.5, 1.0 Hz, 1H), 7.52 (d, 7.5 Hz, 1H), 7.31 (m, 1H), 7.26 (m, 1H), 6.72 (d, *J* = 1.0 Hz, 1H), 6.63 (d, *J* = 2.6 Hz, 1H), 6.48 (d, *J* = 2.6 Hz, 1H), 6.31 (s, 1H), 5.75 (s, 1H), 3.92 (m, 1H), 2.11 (m, 2H), 1.83 (s, 3H), 1.73 (m, 1H), 1.47 (m, 1H), 0.73 (d, *J* = 7.0 Hz, 3H), 0.23 (d, *J* = 7.0 Hz, 3H). ^13^C NMR (126 MHz, CDCl_3_) δ 157.1, 156.3, 154.8, 154.7, 140.4, 133.3, 128.7, 124.6, 124.1, 122.8, 121.3, 120.84, 111.2, 109.4, 105.6, 77.3, 77.0, 76.7, 43.1, 39.5, 30.7, 27.6, 23.7, 21.9, 21.6, 15.7. HRMS: *m*/*z* calcd for C_24_H_25_O_3_ [M − H]^−^: 361.1809; found: 361.1816.

#### 4.3.3. Synthesis of Compounds **3**, **4**, and **5**

(−)*-p*-Mentha-2,8-diene-1-ol [(1*R*,4*S*)-1-methyl-4-(prop-1-en-2-yl)cyclohex-2-enol] (72.96 mg, 0.48 mmoL) (Asta Tech, Bristol, PA, USA) was added to a mixture of stemofuran A (100 mg, 0.44 mmoL) and PTSA (25.17 mg, 0.13 mmoL) in benzene (3 mL) at RT and the reaction mixture was stirred for one hour. The reaction mixture was quenched with aqueous NaHCO_3_ and extracted with CH_2_Cl_2._ The solvent was removed in vacuo and the residue (185 mg) was purified over a CPTLC (2 mm silica gel P254 disc, 45 RPM), and elution with (1% to 10%) EtOAc in hexane yielded a fraction rich with compounds **3** (13 mg, 7%) and **5** (75 mg, 48%), and a fraction rich with **4** (48 mg). The fraction rich with **4** was further purified semipreparative RP-HPLC (column: ODS 10 µ, 250 × 10 mm; detector; UV 254 nm), using 95% MeCN-H_2_O as solvent to afford pure compound **4** (12 mg, 2.7%).

#### 4.3.4. Compound **3**

(1′*S*,2′*S*)-4-(Benzofuran-2-yl)-5′-methyl-2′-(prop-1-en-2-yl)-1′,2′,3′,4′-terahydro-[1,1′ -biphenyl]-2,6-diol; Gum, [α]^26^_D_ +79 (c 0.33, CDCl_3_),^1^H NMR (500 MHz, CDCl_3_) δ 7.58 (dd, *J* = 7.7, 1.2 Hz, 1H), 7.50 (d, 8.1 Hz, 1H), 7.27 (m, 1H), 6.95 (s, 1H), 5.62 (bs, 1H), 4.70 (s, 1H), 4.61 (s, 1H), 4.60 (s, 1H), 3.96 (m, 1H). 2.50 (ddd, *J* = 14.6, 11.3, 4.3 Hz, 1H), 2.30 (m, 1H), 2.16 (m, 1H), 1.89–1.83 (m, 2H), 1.85 (s, 3H), 1.72 (s, 3H). ^13^C NMR (126 MHz, CDCl_3_) δ 155.5 × 2, 154.8 × 2, 148.8, 140.5, 129.9, 129.2, 124.2, 123.5, 122.8, 120.8, 117.5, 111.2, 111.1 × 2, 101.4 × 2, 46.1, 37.3, 30.4, 28.4, 23.6, 20.2. HRMS: *m*/*z* calcd for C_24_H_23_O_3_ [M − H]^−^: 359.1653; found: 359 1653.

#### 4.3.5. Compound **4**

(1*R*,1′′*S*,2*R*,2′′*S*)-2′-(Benzofuran-2-yl)-5,5′′-dimethyl-2,2′′-di (prop-1-en-2-yl)1,1′′,2,2′′, 3,3′′,4,4′′-octahydro [1,1′,3′,1′′-terphenyl]-4′6′-diol; Gum, [α]^26^_D_ 168 (c 0.33, CDCl_3_),^1^H NMR (500 MHz, CDCl_3_) δ 7.58 (d, *J* = 7.5 Hz, 1H), 7.46 (d, *J* = 7.8 Hz, 1H), 7.26 (m, 1H), 7.22 (m, 1H), 6.53 (s, 1H), 6.42 (s, 1H), 5.71 (bs, 1H), 5.62 (bs,1H), 4.52 (s, 2H), 4.17 (d, *J*=6.7 Hz, 2H), 3.53 (s, 1H), 2.92 (s, 1H), 2.15 (s, 2H), 1.98 (m, 2H), 1.80 (s, 6H), 1.63 (m, 2H), 1.44 (m, 2H), 1.20 (s,3H), 1.08 (s,3H) ^13^C NMR (126 MHz, CDCl_3_) δ 154.8, 154.6, 154.5, 146.7, 139.6, 139.1, 132.9, 128.6, 124.7, 124.5, 123.7, 122.6, 121.8, 120.6, 111.7, 111.5, 111.3, 108.1, 107.1, 77.3, 77.0, 76.8, 45.5, 45.2, 42.1, 41.7, 30.3, 30.1, 29.7, 27.9, 27.6, 23.6, 19.9, 19.3. HRMS: *m*/*z* calcd for C_34_H_37_O_3_ [M − H]^−^: 493.2748; found: 493.2752.

#### 4.3.6. Compound **5**

(1*S*,2′*S*)-6′-(Benzofuran-2-yl)-5′-methyl-2′-(prop-1-en-2-yl)1,2′,3′4′-tetrahydro-[1,1′-biphenyl]-2,4-diol; Gum, [α]^26^_D_ 82 (c 0.33, CDCl_3_),^1^H NMR (500 MHz, CDCl_3_) δ 7.61 (dd, *J* = 7.7, 1.4Hz, 1H), 7.53 (d, *J* = 8.1Hz, 1H), 7.52 (d, *J* = 7.8 Hz, 1H), 7.28 (m, 2H), 6.68 (d, *J* = 1.0 Hz, 1H), 6.60 (d, *J* = 2.6 Hz, 1H), 6.48 (d, *J* = 2.6 Hz, 1H), 6.32 (s, 1H), 5.80 (bs, 1H), 5.05 (bs, 1H), 4.47 (bs, 1H), 4.25 (bs, 1H), 4.0 (m, 1H), 2.54 (td, *J* = 10.0, 2.0 Hz, 1H), 2.25 (m, 1H), 2.11 (bs, 1H), 1.86 (s, 3H), 1.75 (m, 1H), 1.61 (m, 1H), 1.16 (s,3H). ^13^C NMR (126 MHz, CDCl_3_) δ 157.0, 156.3, 154.7, 154.7, 146.8, 140.4, 133.1, 128.8, 124.1, 122.8, 120.9, 120.7, 111.8, 111.3, 109.3, 105.6, 105.3, 45.9, 40.4, 30.3, 27.9, 23.7, 19.2. HRMS: m/z calcd for C_24_H_23_O_3_ [M − H]^−^: 359.1653; found: 359.1648.

#### 4.3.7. Synthesis of Compounds **6**–**11**

(−)*-p*-Mentha-2,8-diene-1-ol (72.96 mg, 0.48 mmoL) was added to a mixture of pinosilvin (100 mg, 0.44 mmoL) and PTSA (25.17 mg, 0.13 mmoL) in benzene (3 mL) at RT and the reaction mixture was stirred for one hour. The reaction mixture was quenched with aqueous NaHCO_3_ and extracted with CH_2_Cl_2._ The solvent was removed in vacuo and the residue (185 mg) was separated over a CPTLC (2 mm, silica gel, P254 disc, 45 RPM), and elution with EtOAc in hexane (1% to 10%) yielded a fraction rich with compound **6** (45 mg), **7** (42 mg, 28%), **8** (15 mg, 10%), a (1:0.2) mixture of cyclized products **9** and **10** (18 mg, 12%), and **11** (100 mg, 66%). The fraction rich with compound **6** was further purified by semipreparative RP-HPLC, using 80% MeCN-H_2_O as solvent to afford **6** (15 mg).

#### 4.3.8. Compound **6**

(1*S*, 1′′*S*, 2*S*, 2′′*S*)-5,5′′-Dimethyl-2,2′′-di(prop-1-en-2-yl)-6′-(E-styryl)-1,1′′,2,2′′, 3,3′′,4,4′′-octahydro-[1,1′:3′1′′-terpenyl]-2′,4′-diol; Gum, [α]^26^_D_ 67 (c 0.33, CDCl_3_),^1^H NMR (500 MHz, CDCl_3_) δ 7.42 (d, *J* = 8.0 Hz, 2H), 7.35 (t, *J* = 7.4 Hz, 2H), 7.26 (t, *J* = 7.7 Hz, 2H), 7.25 (d, *J* = 15.8 Hz, 1H), 6.82 (d, *J* = 15.8 Hz, 1H), 6.60 (s,1H), 5.59 (d, *J* = 7.6 Hz, 2H), 4.62 (s, 1H), 4.53 (s, 1H) 4.48 (d, *J* = 4.8 Hz, 2H), 4.05 (d, *J* = 12.0 Hz, 1H), 3.78 (d, *J* = 12.0 Hz, 1H), 2.52 (m, 1H), 2.47 (m, 1H), 2.30–2.14 (m, 4H), 1.84 (m, overlap,4H), 1.82 (s, 6H), 1.75 (s, 3H), 1.53 (s, 3H). ^13^C NMR (126 MHz, CDCl_3_) δ 154.4, 153.8, 147.5, 140.3, 139.5, 137.9, 137.1, 130.7, 128.7, 128.6 × 2, 127.9, 127.3, 126.4 × 2, 124.5 × 2, 119.2, 117.5, 111.7, 111.4, 106.8, 46.5, 44.9, 40.4, 35.9, 30.4, 30.4, 28.4, 28.3, 23.7, 23.7, 21.0, 18.9. HRMS: *m*/*z* calcd for C_34_H_41_O_2_ [M + H]^+^: 481.3112; found: 481.3106.

#### 4.3.9. Compound **7**

(1*S*, 1′′*S*, 2*S*, 2′′*S*)-5,5′′-Dimethyl-2,2′′-di(prop-1-en-2-yl)-2′-((*E*)-styryl-1,1′′,2,2′′,3,3′′, 4,4′′-octahydro-[1,1′:3′1′′-terpenyl]-4′,6′-diol; Gum, [α]^26^_D_ 118 (c 0.33, CDCl_3_),^1^H NMR (500 MHz, CDCl_3_) δ 7.40 (dd, *J* = 8.5, 1.0 Hz, 2H), 7.39 (t, *J* = 7.4 Hz, 2H), 7.38 (t, *J* = 5.2 Hz, 1H), 6.83 (d, *J* = 16.5 Hz, 1H), 6.34 (s, 1H), 6.18 (d, *J* = 16.5 Hz, 1H), 5.95 (s, 1H), 5.58 (bs, 1H), 5.29 (bs, 1H), 4.56 (bs, 1H), 4.34 (bs, 1H), 3.72 (m, 1H), 2.50 (m, 2H) 2.18 (m, 2H), 2.00 (m, 2H), 1.77 (s, 6H), 1.43 (s, 6H). ^13^C NMR (126 MHz, CDCl_3_) δ 154.7 × 2, 147.2 × 2, 140.5, 139.2 × 2, 137.3, 134.9, 128.6, 127.4 × 2, 126.1 × 2, 124.9, 119.7, 111.4 × 2, 104.6, 45.4 × 2, 53.4, 41.2 × 2, 30.2, 27.9 × 2, 23.6 × 2, 20.6 × 2. HRMS: *m*/*z* calcd for C_34_H_41_O_2_ [M + H]^+^: 481.3101; found: 481.3104.

#### 4.3.10. Compound **8**

(1′*S*,2′*S*)-5′-Methyl-2′-(prop-1-en-2-yl)-6-(*E*)-styryl)-1′,2′,3′,4′-tetrahydro-[1,1′bipenyl]-2,4-diol. Gum, [α]^26^_D_ 176 (c 0.33, CDCl_3_),^1^H NMR (500 MHz, CDCl_3_) δ 7.50 (d, J= 7.1 Hz, 2H), 7.36 (t, J= 7.1 Hz, 2H), 7.35, 7.28, 7.27 (m, 1H), 7.26, 7.25, 7.06, 7.02 (d, *J* = 15 Hz, 1H), 6.96 (d, *J* = 15 Hz, 1H), 6.65 (d, brs, 1H), 6.54 (d, brs, 1H), 5.60 (s, 1H), 4.70 (s, 1H), 4.60 (s, 1H), 3.91 (m, 1H), 2.46 (m, 1H), 2.15 (s, 1H), 2.25 (m, 1H), 1.84 (s, 3H), 1.71 (s, 3H). ^13^C NMR (126 MHz, CDCl_3_) δ 156.6, 154.3, 149.1, 140.4, 137.2, 137.10, 128.6 × 2, 128.1, 127.5, 126.4 × 2, 123.7, 116.4, 111.1, 108.1, 106.0, 46.2, 37.3, 30.4, 28.4, 23.7, 20.4. HRMS: *m/z* calcd for C_24_H_27_O_2_ [M + H]^+^: 347.2006; found: 347.2019.

#### 4.3.11. Compounds **9**+**10**

Compound **9**: (6*S*,10*S*)-6,6,9-Trimethyl-3-[(*E*)-styryl]-6,7,8,10-tetrahydro-6*H*-benzo [c]chromene-1-ol; Gum, ^1^H NMR (500 MHz, CDCl_3_) δ 7.55 (d, *J* = 7.1 Hz, 2H), 7.39 (m, 2H), 7.30 (m, 1H), 7.21 d, *J* = 15 Hz, 1H), 7.02 (d, *J* = 15 Hz, 1H), 6.75 (d, *J* = 2.6 Hz, 1H), 6.30 (d, *J* = 2.6 Hz, 1H), 5.88 (m, 2H), 3.36 (m, 1H), 2.22 (m, 2H), 1.94 (m, 1H), 1.74 (m, 1H), 1.62 (s, 3H), 1.47 (s, 3H, 1H), 1.15 (s, 3H), 1.12 (s, 3H). ^13^C NMR (126 MHz, CDCl_3_) δ 155.0, 154.7, 137.6, 137.5, 134.4, 128.7 × 2, 128.6, 128.0, 127.6, 126.6 × 2, 126.2, 116.2, 105.7, 103.7, 77.2, 46.4, 34.5, 31.1, 27.5, 25.2, 23.2, 18.9.

Compound **10**: (5*S*, 6*S*)-2-Methyl-5-(prop-1-en-2-yl)-9-styryl-3,4,5,6-tetrahydro-2*H*-2,6 -methanobenzo[b]-oxocin-7-ol; Gum, ^1^H NMR (400 MHz, CDCl_3_) 7.51 (m, 2H), 7.36 (m, 2H), 7.29 (m, 1H), 7.21 (d, *J* = 15 Hz, 1H), 7.02 (d, *J* = 15 Hz, 1H), 6.33 (d, *J* = 2.6 Hz, 1H), 6.71 (d, *J* = 2.6 Hz, 1H), 5.04 (m, 1H), 4.99 (s, 1H), 3.5 (bs, 1H), 2.33 (bs, 1H), 1.61 (s, 3H), 1.16 (s, 3H). δ ^13^C NMR (101 MHz, CDCl_3_) δ 157.6, 154.7, 145.6, 137.4, 136.4, 130.9, 128.8 × 2, 127.8, 126.5 × 2, 124.6, 117.3, 111.4, 103.8, 102.1, 74.6, 44.05, 35.3, 30.7, 30, 29.4, 22.8, 20.7. HRMS: *m*/*z* calcd for C_24_H_25_O_2_ [M − H]^−^: 345.1860; found: 345.1857.

#### 4.3.12. Compound **11**

(1′*S*,2′*S*)-5′-Methyl-2′-(prop-1-en-2-yl)-4-[(*E*)-styryl-1′,2′,3′,4′-tetrahydro-[1,1′-biphenyl]-2,6-diol; Gum, [α]^26^_D_ 0 (c 0.33, CDCl_3_), ^1^H NMR (500 MHz, CDCl_3_) δ 7.43 (d, *J* = 8.6 Hz, 2H), 7.34 (t, *J* = 7.5 Hz, 2H), 7.25 (d, *J* = 12.0 Hz, 1H), 7.23 (d, *J* = 16.0 Hz, 1H), 6.75 (d, *J* = 16.0 Hz, 1H), 6.57 (d, *J* = 2.0 Hz, 1H), 6.33 (d, *J* = 2.0 Hz, 1H), 5.6 (s, 1H), 4.64 (bs, 1H), 4.50 (bs, 1H), 3.77 (m, 1H), 2.24 (m, 1H), 2.12 (m, 1H), 1.82 (m, 2H), 1.80 (s, 3H), 1.8 (s, 3H), 1.54 (s, 3H). ^13^C NMR (126 MHz, CDCl_3_) δ 156.4, 154.8, 147.2, 140.3, 139.7, 137.5, 131.4, 128.6 × 2, 127.8, 127.5, 126.4 × 2, 124.2, 120.1, 111.7, 105.8, 103.9, 45.5, 39.9, 30.2, 30.0, 23.7, 20.9. HRMS: m/z calcd for C_24_H_25_O_2_ [M − H]^−^: 345.1860; found: 345.1861.

#### 4.3.13. Synthesis of Compound **12**–**14**

Geraniol (158.56 mg, 1.02 mmoL) was added to a mixture of pinosilvin (200 mg, 0.94 mmoL) and PTSA (53.8 mg, 0.26 mmoL) in benzene (5 mL) at RT and the reaction mixture was stirred for one hour. The reaction mixture was quenched with aqueous NaHCO_3_ and extracted with CH_2_Cl_2._ The solvent was removed in vacuo and the residue (265 mg) was purified over a CPTLC (2 mm silica gel P254 disc, 45 RPM), and elution with (1% to 10%) EtOAc in hexane yielded compounds **12** (42 mg, 9%), **13** (72 mg, 22%), and **14** (50 mg, 15%).

#### 4.3.14. Compound **12**

2,4-bis[(*E*)-3,7-Dimethylocta-2,6-dien-1-yl]-5-(*E*)-styryl)benzene-1,3-diol; ^1^H NMR (400 MHz, CDCl_3_) δ 7.51 (dd, *J* = 7.0, 1.6 Hz, 2H), 7.38 (td, *J* = 7.2, 1.7 Hz, 2H), 7.31 (d, *J* = 16.0 Hz, 1H), 7.28 (tt, 6.6, 1.4 Hz, 1H), 6.93 (d, *J* = 16.0 Hz, 1H), 6.73 (s, 1H), 5.32 (m, 1H), 5.25 (m, 1H), 5.10 (m, 2H), 3.48 (bd, *J*= 4.6 Hz, 4H), 2.14 (m, 4H), 2.10 (m, 4H), 1.86 (s, 6H), 1.64 (s, 6H). ^13^C NMR (101 MHz, CDCl_3_) δ 153.7, 153.1, 139.0, 137.6, 137.4, 135.6, 132.0, 131.9, 130.4, 128.7 × 2, 127.5 × 2, 126.6, 126.5 × 2, 123.9, 123.8, 122.7, 121.6, 118.2, 113.8, 105.5, 39.8, 39.7, 26.5, 26.4, 25.73, 25.70, 22.8, 17.74, 17.73, 16.3, 16.2. HRMS: *m*/*z* calcd for C_34_H_43_O_2_ [M − H]^−^: 483.3269; found: 483.3264.

#### 4.3.15. Compound **13**

2-[(*E*)-3,7-Dimethylocta-2,6-dien-1-yl]-5-[(*E*)-styryl)benzene-1,3-diol; Gum, ^1^H NMR (400 MHz, CDCl_3_) δ 7.51 (dd, *J* = 7.0, 1.6 Hz, 2H), 7.36 (td, *J* = 7.2, 1.7 Hz, 2H), 7.28 (tt, 6.6, 1.2 Hz, 1H), 7.10 (d, *J* = 16 Hz, 1H), 6.97 (d, *J* = 16 Hz, 1H), 6.62 (s, 2H), 5.33 (m, 1H), 5.09 (m, 1H), 3.48 (d, *J* = 7.0 Hz, Hz, 2H), 2.09–2.13 (m, 4H), 1.86 (s, 3H),1.73 (s, 3H), 1.64 (s, 3H). ^13^C NMR (101 MHz, CDCl_3_) δ 155.2, 139.4, 137.2, 136.8, 132.1, 128.7 × 2, 128.6, 128.1, 127.6, 126.5 × 2, 123.7, 121.3, 113.4, 106.5 × 2, 39.7, 26.4, 25.7, 22.5, 17.7, 16.3. HRMS: *m*/*z* calcd for C_24_H_27_O_2_ [M − H]^−^: 347.2017; found: 347.2013. NMR data of this compound agreed with those reported for amorphastibol [25].

#### 4.3.16. Compound **14**

4-[(*E*)-3,7-Dimethylocta-2,6-diene-1-yl)-5-[(*E*)-styryl]benzene-1,3-diol; Gum, ^1^H NMR (400 MHz, CDCl_3_) δ 7.52 (dd, *J* = 7.0, 1.6 Hz, 2H), 7.39 (td, *J* = 7.2, 1.7 Hz, 2H), 7.37 (m, 1H), 7.32 (d, *J* = 16 Hz, 1H), 6.95 (d, *J* = 16 Hz, 1H), 6.70 (d, *J* = 2.5 Hz, 1H), 6.34 (d, *J* = 2.5 Hz, 1H), 5.21 (m, 1H), 5.07 (m, 1H), 3.47 (d, 6.7 Hz, 2H), 2.09 (m, 2H), 2.07 (m, 2H), 1.85 (s, 3H), 1.68 (s, 3H), 1.60 (s, 3H). ^13^C NMR (101 MHz, CDCl_3_) δ 155.6, 154.5, 138.6, 137.7, 137.4, 131.9, 131.2, 128.7, 127.7, 126.6, 126.5, 123.9, 122.3, 118.0, 105.5, 102.9, 39.7, 26.5, 25.7, 25.0, 17.7, 16.3. HRMS: *m*/*z* calcd for C_24_H_28_O_2_ [M − H]^−^: 347.2017; found: 347.2018.

#### 4.3.17. Synthesis of Compound **15** and **16**

(−)*-p*-Mentha-2,8-diene-1-ol (290.3 mg, 1.9 mmoL) was added to a mixture of resveratrol (400 mg, 1.75 mmoL) and PTSA (99.9 mg, 0.52 mmoL) in benzene (5 mL) at RT and the reaction mixture was stirred for 2 h. The reaction mixture was quenched with aqueous NaHCO_3_ and extracted with CH_2_Cl_2._ The solvent was removed in vacuo and the residue (420 mg) was purified over a CPTLC (2 mm silica gel P254 disc, 45 RPM), and elution with (1% to 20%) EtOAc in hexane gave unreacted resveratrol, compounds **15** (27 mg, 3%) and **16** (9 mg, 1%).

#### 4.3.18. Compound **15**

(1*S*, 1′′*S*, 2*S*, 2′′*S*)-2′-((*E*)-4-Hydroxystyryl)-5,5′′-dimethyl-2,2′′-di(prop-1-en-2-yl) -1,1′′,2,2′′, 3,3′′,4,4′′-octahydro-[1,1′:3′1′′-terphenyl]-4′,6′-diol; Gum, [α]^26^_D_ 0 (c 0.33, CDCl_3_);^1^H NMR (400 MHz, CDCl_3_) δ 7.30 (d, *J*= 7.8 Hz, 2H), 6.83 (d, *J* = 6.6 Hz, 2H) 6.65 (d, *J* = 16.5 Hz, 1H), 6.35 (bs, 1H), 6.08 (d, *J* = 16.5 Hz, 1H), 5.61 (bs, 2H), 4.57 (s, 2H), 3.76 (d, *J* = 10 Hz, 2H 2.52 (m, 2H), 2.08–2.02 (m, 2H), 1.79 (s, 6H), 1.75 (m, 4H), 1.45 (s, 6H). ^13^C NMR (101 MHz, CDCl_3_) δ 155.3, 154.6 × 2, 147.2 × 2, 140.8, 139.1 × 2, 134.4, 130.3, 127.5 × 2. 126.4, 125.1 × 2, 119.9 × 2, 115.6 × 2, 111.4 × 2, 104.5, 45.4 × 2, 41.2 × 2, 30.2 × 2, 28.0 × 2, 23.6 × 2, 20.5 × 2. HRMS: *m*/*z* calcd for C_34_H_39_O_3_ [M − H]^−^: 495.2905; found: 495.2905.

#### 4.3.19. Compound **16**

(1*S*, 1′′*S*, 2*S*, 2′′*S*)-6′-((*E*)-4-hydroxystyryl)-5,5′′-dimethyl-2,2′′-di(prop-1-en-2-yl) -1,1′′,2,2′′, 3,3′′,4,4′′-octahydro-[1,1′:3′1′′-terphenyl]-2′,4′-diol; Gum, [α]^26^_D_ 155 (c 0.33, CDCl_3_); ^1^H NMR (400 MHz, CDCl_3_) δ 7.34 (d, *J* = &.8 Hz, 2H), δ 7.11 (d, *J* = 16.0 Hz, 1H), 6.83 (d, *J* = 8.6 Hz, 2H), 6.73 (d, *J* = 16.0 Hz, 1H), 6.6 (s, 1H), 4.56 (bs, 1H), 4.61 (bs, 1H), 4.5 (d, *J* = 6.2 Hz, 1H), 4.04 (d, *J* = 10 Hz, 1H), 3.77 (d, *J* = 10 Hz, 1H), 2.5 (m, 2H), 2.26 (bs, 2H), 1.80 (s, 6H), 1.74 (s, 3H), 1.52 (s, 3H). ^13^C NMR (126 MHz, CDCl_3_) δ 155.1, 154.3, 153.7, 147.5, 147.4, 140.2, 139.5, 137.4, 130.8, 130.2, 127.7 × 2, 125.9, 124.5, 119.1, 117.2, 115.5 × 2, 111.7, 111.4, 106.6, 46.5, 44.9, 40.3, 35.9, 30.58, 30.4, 28.4, 28.2, 23.7, 23.7, 21.01, 14.20. HRMS: *m*/*z* calcd for C_34_H_41_O_3_ [M + H]^+^: 497.3050; found: 497.3055

#### 4.3.20. Synthesis of Compound **17**

Compound **5** (45 mg, 0.12 mmoL) in ethyl acetate was hydrogenated with Adam’s catalyst, PtO_2_ (10%, 4.5 mg) at 40 psi at RT for 24 h, as previously published [40]. The catalyst was removed by filtration and the filtrate was evaporated to obtain a complete hydrogenated product (42 mg). Further purification of the portion of this mixture (20 mg) with RP-HPLC (column: ODS 10µ, 250 × 10 mm; detector; UV 254 nm) using 75% MeCN-H_2_O as solvent to afford 5-(benzofuran-2-yl)-4-((1*R*,2*S*,5*R*)-2-isopropyl-5-methylcyclohexyl)benzene-1,3-diol (**17**, 7.0 mg) as the major compound.

#### 4.3.21. Compound **17**

5-(Benzofuran-2yl)-4-[(1*R*,2*S*,5*R*)-2-isopropyl-5-methylcyclohexyl]benzene-1,3-diol; Gum, [α]^26^_D_ 1 (c 0.33, CDCl_3_),^1^H NMR (400 MHz, CDCl_3_) δ 7.65 ((d, *J* = 7.8 Hz, 1H), 7.52 (d, *J* = 7.8 Hz, 1H), 7.30 (m, 2H) 6.71 (s, 1H), 6.68 (bs, 1H), 6.37 (bs, 1H), 2.91 (m, 1H), 2.13 (m, 1H), 1.76 (m, 1H), 1.05 (m, 1H), 1.0 (d, *J* = 8.0 Hz, 3H), 0.89 (d, *J* = 8.0 Hz, 3H), 0.39 (d, *J* = 8.0 Hz, 3H). ^13^C NMR (101 MHz, CDCl_3_) δ 156.4, 156.1, 154.7, 153.85 133.3, 128.9, 124.0, 123.1, 122.7, 120.9, 111.2, 109.7, 105.2, 104.8, 44.5, 43.1, 40.8, 35.4, 33.3, 28.4, 25.3, 22.7, 21.6, 15.5. HRMS: *m*/*z* calcd for C_24_H_27_O_3_ [M − H]^−^: 363.1966; found: 363.1970, 399.1742 [M + Cl]^−^.

### 4.4. Antimicrobial Assay

Antimicrobial assays were carried out using a published method [10]. All organisms were obtained from the American Type Culture Collection (Manassas, VA, USA). These organisms included *Candida albicans* ATCC 90028, *Cryptococcus neoformans* ATCC 90113, *Aspergillus fumigatus* ATCC 204305, methicillin-resistant *Staphylococcus aureus* ATCC 1708 (MRSA), *Escherichia coli* ATCC 2452, *Pseudomonas aeruginosa* ATCC BAA-2018, *Klebsiella pneumonia* ATCC 2146, and vancomycin resistant *Enterococcus faecium* (VRE) ATCC 700221. Briefly, the antimicrobial activity was determined through a high throughput screening assay performed in a 384-well plate. Appropriate drug controls for bacteria and fungi were included in each assay. The concentration of compound/fraction responsible for 50% growth inhibition (IC_50_) was calculated using XLfit 4.2 software (IDBS, Alameda, CA), with fit model 201. Minimum inhibitory concentration (MIC) was defined as the lowest test concentration that afforded no visual growth. Susceptibility testing was performed using a modified version of the CLSI (formerly NCCLS) method [41,42].

### 4.5. Antimicrobial Combination Study by Checkerboard Method

The combination study of the compounds was carried out in MRSA using the standard checkerboard method by Norden et al. 1979 [43]. Test samples were dissolved in 100% DMSO to the desired concentrations, and serially diluted (1:2) with 20% DMSO/saline. For checkerboard, compound **5**, **17** and machaeriol C (i.e., previously isolated from *Machaerium* sp. [8,9]) were tested in the range of 2.5 µg/mL to 0.039 µg/mL. Inocula was prepared in cation-adjusted Mueller- Hinton broth to afford 5 × 10^5^ colony forming units per mL. Samples were transferred to 96-well assay plates (10 µL) in a checkerboard layout followed by inocula (180 µL). The MIC of each antimicrobial agent alone and in combination was determined after incubation at 35 °C for 24 h. The fractional inhibitory concentration (FIC) was calculated by using the following formula: FIC = [A*]/[A] + [B*]/[B], where [A*] is the MIC of compound A in the presence of compound B, [A] is the MIC of compound A alone, [B*] is the MIC of compound B in the presence of compound A, and [B] is the MIC of compound B alone. FIC: 0.5 = synergistic; 0.51–1.0 = additive; 1.1–2.0 = indifferent; >2.0 = antagonistic [44,45].

### 4.6. Bacterial Cultures for In Vivo Experiment

MRSA USA300 (kindly provided by Jorge Vidal, University of Mississippi Medical Center, Jackson, MS, USA) was isolated on tryptic soy agar (TSA), then grown in culture for 16 h at 37 °C with shaking in tryptic soy broth (TSB). The bacteria were then diluted 100-fold in sterile TSB and grown to mid-logarithmic phase in tryptic soy broth (TSB). Serial dilutions were prepared in sterile PBS to achieve a target inoculum of 10^7^ colony-forming units (CFU) per 0.03 mL.

### 4.7. In Vivo Anti-MRSA Nosocomial Assay in Murine Model

Six- to eight-week-old female C57BL/6J mice (the Jackson Laboratory, Bar Harbor, Maine) were anesthetized with a mixture of ketamine and xylazine and weighed. Each anesthetized mouse received a target inoculum of 10^7^ CFU, as performed by Achouiti et al. [46]. The serial dilution and plating of the inoculum were carried out to verify the purity and accuracy of the dose. The actual dose was determined to be 2 × 10^7^ CFU. Twelve hours after infection, the mice were anesthetized and administered 0.048 mL of vehicle ciprofloxacin (30 mg/Kg), compound **5** (30 mg/Kg), or compound **17** (30 mg/Kg). The vehicle composition, which was also used for suspension of the compounds and dilution of ciprofloxacin, was ethanol:DMSO:Cremophor EL:PBS (5:5:10:80 *v/v*).

Twenty-four hours after infection, the mice were anesthetized, then euthanized with an overdose of sodium pentobarbital. Using a midline incision, each trachea was exposed and another incision was made partially through the middle of the trachea, being careful not to completely sever the trachea all the way through. Nasopharyngeal lavage was performed by placing a pipette into the tracheal incision, pushing 0.05 mL of sterile PBS through the trachea 2 times and into a 1.5 mL tube placed under the nose. The lavage fluid was then diluted and plated on TSA for bacterial load quantitation. Next, the whole lung was removed and homogenized before being plated and counted to determine bacterial loads.

### 4.8. Transcriptional Reporter Assays

T98G glioblastoma multiforme cells from ATCC were plated in white opaque 384-well plates at a density of 4300 cells/well in 30 μL of growth medium (DMEM with 10% FBS and 1% Pen/Strep). On the next day, the medium was aspirated and replaced with DMEM containing 10% FBS. The cells were transfected with respective plasmids using X-tremeGENE HP transfection reagent (Roche). The luciferase vectors used in this assay are summarized in (Appendix A). After 24 h of transfection, the test agents were added to the transfected cells, followed 30 min later by an inducing agent (IL-6 for Stat 3, TGF-β for Smad, m-wnt3a for Wnt and PMA for AP-1, NFkB, E2F, Myc, ETS and Hedgehog). No inducer was added for FoxO, miR-21, Ras, AhR and pTK vector control. After 4 h or 6 h of induction, the cells were lysed by the addition of a One-Glo luciferase assay system (Promega, Madison, WI, USA). The light output was detected in a Glomax Multi+ detection system with Instinct Software (Promega). This luciferase assay determined whether the test agent was able to inhibit the activation of the cancer-related signaling pathways. In the case of FoxO, mi-R21-, Ras-, and AhR-enhanced luciferase activity by the test agents was assessed [47].

## Data Availability

Data provided in Appendix A.

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
