# Peer review of "Synthesis and Inhibitory Activity of Machaeridiol-Based Novel Anti-MRSA and Anti-VRE Compounds and Their Profiling for Cancer-Related Signaling Pathways"

_molecules, 2022, doi:10.3390/molecules27196604_

Round 1

Reviewer 2 Report

Recommendation: Accept after minor revisions

Comments:

The submitted manuscript entitled “Synthesis and Inhibitory Activity of Machaeridiol- based Novel anti-MRSA and anti-VRE Compounds and their Profiling for Cancer- Related Signaling Pathways” evaluated the anti-MRSA, anti-VRE activities and the profiling of cancer -related signaling pathways of 17 synthetic analogues of machaeriols A-D and machaeridiols A-C. This paper is of interest to the readers of Molecules. I would recommend accepting the manuscript after the following comments are addressed.

General comments:

Generally, there should be a space between values and units. Please check and correct them throughout the whole manuscript.

Moderate English changes are required.

There are inconsistencies in the format of the reference session. Just keep the volume number and remove the issue number if applied. Please check through the whole session carefully.

Corrections:

Generally, grammar mistakes, eg: machaeriols A-D; machaeridiols A-C; activities

For the detailed pieces of the corrections, please check the attached highlighted version.

L111: The word “skeleton” is preferred instead of “nucleus”.

L112: Please rephrase the sentence.

L118: “moiety”

L161: “Among the analogues of …”

L328: Please double check the spelling of “anizaldehyde reagent”

L333: This sentence should be removed from this purification session

Round 2

Reviewer 1 Report

The manuscript entitled “Synthesis and Inhibitory Activity of Machaeridiol- based Novel 2 anti-MRSA and anti-VRE Compounds and their Profiling for 3 Cancer- Related Signaling Pathways” recommended for acceptance in Molecules journal.